# Developing Non-Human Primate Models of Inherited Retinal Diseases

**DOI:** 10.3390/genes13020344

**Published:** 2022-02-14

**Authors:** Ivan Seah, Debbie Goh, Hwei Wuen Chan, Xinyi Su

**Affiliations:** 1Department of Ophthalmology, Yong Loo Lin School of Medicine, National University of Singapore, 1E Kent Ridge Road, NUHS Tower Block Level 7, Singapore 119 228, Singapore; ivanseah@nus.edu.sg (I.S.); hwei_wuen_chan@nuhs.edu.sg (H.W.C.); 2Department of Ophthalmology, National University Hospital, 1E Kent Ridge Road, NUHS Tower Block Level 7, Singapore 119 228, Singapore; debbie.goh@mohh.com.sg; 3Institute of Molecular and Cell Biology (IMCB), Agency for Science, Technology and Research (A*STAR), 61 Biopolis Drive, Singapore 138 673, Singapore; 4Singapore Eye Research Institute (SERI), The Academia, 20 College Road, Level 6 Discovery Tower, Singapore 169 856, Singapore

**Keywords:** hereditary eye diseases, retinitis pigmentosa, stargardt disease, leber congenital amaurosis

## Abstract

Inherited retinal diseases (IRDs) represent a genetically and clinically heterogenous group of diseases that can eventually lead to blindness. Advances in sequencing technologies have resulted in better molecular characterization and genotype–phenotype correlation of IRDs. This has fueled research into therapeutic development over the recent years. Animal models are required for pre-clinical efficacy assessment. Non-human primates (NHP) are ideal due to the anatomical and genetic similarities shared with humans. However, developing NHP disease to recapitulate the disease phenotype for specific IRDs may be challenging from both technical and cost perspectives. This review discusses the currently available NHP IRD models and the methods used for development, with a particular focus on gene-editing technologies.

## 1. Introduction

Inherited retinal diseases (IRDs) represent a genetically and clinically heterogenous group of diseases in which genetic mutations critical to retinal function lead to photoreceptor cell and/or retinal pigmented epithelium (RPE) death and, consequently, progressive visual loss. To date, over 260 disease genes have been identified (“RetNet, the Retinal Information Network”) [1]. Exon sequencing of these genes has identified the genetic cause in approximately two-thirds of IRD patients [2]. Next-generation sequencing technologies are expected to facilitate further identification of novel disease-causing genes, non-coding mutations and structural variants in the genome.

The identification of disease-contributing genetic mutations has led to a better understanding of IRD pathogenesis, culminating in efforts toward therapeutic development. In 2017, the United States Food and Drug Administration (FDA) approved Voretigene Neparvovec (Luxturna), an adeno-associated virus (AAV) vector-based therapy for patients with biallelic *RPE65* mutation-associated Leber congenital amaurosis (LCA). While such use-cases are still extremely rare, with predictions of approximately 15,000 eligible patients globally [3], the event highlighted the possibility of developing therapeutics for treating more prevalent mutations such as *ABCA4*-, *USH2A*- and *EYS*-associated IRDs [4].

Due to the enormous costs associated with clinical trials in therapeutic development and the scarcity of IRD patients, the pre-clinical development of gene therapeutics is of utmost importance. An animal model that recapitulates the pathogenesis of human IRDs is a key asset in this process. Over the past two decades, over 200 animal models of IRDs have been identified [5]. These models have been valuable resources for studying vision physiology and IRD pathogenesis. Of these models, non-human primates (NHP) have the greatest utility as they have the most anatomically and physiologically similar retina to humans.

However, developing NHP models can be challenging. While attempts have been made to develop or identify naturally occurring models, there is still a significant paucity of such models. Identification or creation of NHP IRD models can also be extremely resource-intensive as it requires significant validation of not only the phenotype but the genotype as well. This review discusses the available IRD NHP models and the methods used for their development. These include naturally occurring models, models created using laser or chemical methods and models created with genetic methods (Table 1).

## 2. The Importance of Genotype–Phenotype Correlation in IRDs and Implications for NHP Disease Modelling

The process of diagnosing IRDs has transformed rapidly over the years. Historically, IRDs were described phenotypically. This was based on establishing symptom chronology and a detailed pedigree revealing familial inheritance. Subsequently, further structural and functional assessments were included. Structural investigations include fundus photography (FP), autofluorescence imaging (AF), optical coherence tomography (OCT) and their wide-field counterparts. Functional assessments include visual field testing and electroretinogram (ERG). To date, there are more than 20 IRD phenotypes, with four major types which are most readily recognized clinically: (1) rod–cone degenerations (e.g., Retinitis pigmentosa (RP)), (2) cone–rod degenerations (e.g., Achromatopsia), (3) chorioretinal degenerations (e.g., Choroideremia, gyrate atrophy) and (4) inherited macular dystrophies (e.g., Stargardt Disease, X-linked Retinoschisis).

However, diagnosing specific IRDs based on clinical presentation can be challenging due to genetic and phenotypic heterogeneity. To date, over 100 genes have been implicated to cause RP, the most commonly diagnosed IRD, where rod photoreceptor cells loss precede that of cones. Although the age of onset, pedigree and disease severity may provide certain clues, marked phenotypic overlap, even with other IRDs such as advanced cone–rod dystrophy, often precludes accurate diagnosis. Furthermore, many factors determining disease–gene penetrance and expression are not fully elucidated, resulting in phenotypical heterogeneity for identified mutations. Finally, in the late stages of many IRDs, the retina is extremely atrophic with widespread retinal cell death, making it almost impossible to identify the underlying disease.

Hence, genotype–phenotype correlation is increasingly recognized to be crucial for IRD diagnoses these days. The declining costs of sequencing have also made genetic testing feasible. Today, genetic testing is fundamental towards IRD diagnosis as it improves the accuracy of diagnosis and prognosis; provides patients and families with specific inheritance risks; facilitates pre-implantation genetics; and, most importantly, allows patients access to FDA-approved gene therapeutics and clinical trials.

The paradigm shift towards recognizing the genotype as the basis for disease development also has implications on animal model design. With an increased regulatory emphasis on testing therapeutics in biologically relevant models, an IRD model for therapeutic testing should ideally capture disease pathogenesis by showcasing both the disease genotype and phenotype. Furthermore, the model should also allow investigators to understand the systemic effects of the retinal therapeutic and associated challenges with the mode of therapeutic delivery. Given these criteria, transgenic NHP IRD models will likely be the gold-standard in IRD disease modelling.

## 3. The Non-Human Primate in Ophthalmology Disease Modelling

NHP models have been widely used in biomedical research due to their shared similarities with humans in terms of anatomy, physiology, genetics and embryology. Three main NHPs are commonly used for human disease modelling. These are the *Macaca fascicularis* (crab-eating macaque or cynomolgus), *Macaca mulatta* (rhesus monkey) and *Callithrix jacchus* (common marmoset).

Even though evolutionary divergence occurred almost 25 million years ago, NHPs and humans still share up to 98.77% DNA sequences in the genome [16]. Hence, many disease susceptibility genes for ophthalmic conditions are also shared, including age-related macular degeneration [17] and retinoblastoma [18]. Given the likelihood of sharing similar pathogenic mechanisms, in disease states with available NHP models, they are considered the gold standard for therapeutic testing. In ophthalmology, NHPs have been routinely used for modelling conditions, including glaucoma, diabetic retinopathy and age-related macular degeneration [19,20,21].

Anatomically, the NHP eye has the unique characteristic of having a macula, an area concentrated with cone photoreceptor cells. Together with the large cortical representation of the central visual field, this accounts for the high-resolution central visual acuity that is shared by NHP and humans. This is not seen in any other animal models to date and has significant implications for studying disease behavior and therapeutic effects.

While other IRD models exist, such as the *RPE65*-associated LCA canine model used in the development of voretigene neparvovec (Luxturna, Novartis AG), the traits mentioned above make the NHP uniquely suitable for studying certain IRDs. For instance, macular dystrophies such as Stargardt disease, Best disease and *RDH5*-related fundus albipunctatus may be better studied in the NHP eye, which has a macula. In comparison, canines only have a bouquet of cones [22]. Furthermore, genetic similarity is also extremely important. Specific genes that contribute to IRD development such as *EYS* for RP may not be present in the murine genome, making modelling such diseases in these small animals challenging.

## 4. Naturally Occurring IRD NHP Models

Naturally occurring IRD NHP models for RP and achromatopsia have been identified in recent years. These were identified through large cohort ophthalmic screens or genetic screens. In one of the RP models, no human-IRD causative gene was identified through genotyping. In a separate model, *BBS7*, a gene implicated in the development of syndromic RP, was identified. Finally, an achromatopsia model with *PDE6C* mutation was also identified recently.

In 2018, Ikeda et al. discovered a *M. fascicularis* family with RP. Through an ophthalmic screen of 1443 monkeys, two genetically related monkeys were found with phenotypic findings suggestive of RP. The index case was a 14-year old, female monkey with significant retinal degeneration and cystoid macular oedema in both eyes. OCT showed the loss of outer retinal layers in the parafoveal region, and full-field ERG (ffERG) was completely unrecordable, suggestive of a late-stage retinal degeneration. Her nephew, a 3-year old, male monkey, demonstrated less severe retinal changes. In both eyes, the loss of outer retinal layers was only noted in the peripheral retina with preservation of the fovea. Scotopic ERG was almost unrecordable, and photopic ERG was extremely reduced. These findings were typical of a rod–cone degeneration, which most RP cases present clinically. Assuming an autosomal recessive mode of inheritance, the team conducted whole-exome sequencing of the family of monkeys (two affected, seven unaffected). However, no conclusive pathogenic mutation could be identified from the analysis [6]. While the model demonstrated phenotypic changes that were suggestive of a retinal degeneration, the inability to identify a human IRD causative mutation limits the applicability of this model for the study of human IRDs.

RP can occur in isolation or as part of several syndromic conditions. RP-associated syndromes include Usher syndrome and Bardet–Biedl syndrome [7,23]. In 2019, Peterson et al. identified a lineage of *M. mulatta* with *BBS7* mutation. Mutations in *BBS7* are associated with the development of Bardet–Biedl syndrome in humans. Bardet–Biedl syndrome is a ciliopathy characterized by retinal degeneration, kidney dysfunction, obesity, hypogonadism and polydactyly [24]. In the report, three monkeys that had a frameshift mutation in exon 3 of the *BBS7* gene c.160delG (p.Ala54fs) manifested varying degrees of findings that were synonymous with syndromic RP and renal impairment. OCT imaging showed retinal thinning with loss of distinct layers most severe at the macula, and AF imaging showed diffuse hyper AF with hypo AF signal at the macula. Histopathology revealed the loss of photoreceptors and RPE, which was most severe centrally. Light-adapted ERG was only done for one monkey but showed almost complete loss of both the a- and b-waves. Two monkeys also had smaller reproductive organs [7]. A human-IRD causative mutation was identified in this model. However, the monkeys were identified after substantial disease progression. Future studies will seek to observe subsequent generations to understand the disease progression in these animals.

Moshiri et al., in 2019, identified four related *M. mulatta* with a homozygous R565Q missense mutation in the catalytic domain of *PDE6C*, a cone-specific phototransduction enzyme associated with achromatopsia in humans. Achromatopsia is a cone disorder caused by mutations in the cone phototransduction machinery. It usually presents bilaterally, affecting all three types of cone photoreceptor cells. The disease can present as rod monochromacy, where there is a total lack of cone function or incomplete where cone function is reduced [25]. In early disease, the fundus can appear normal followed by RPE mottling and atrophy in later stages. Patients are extremely photophobic and have severely impaired color vision. ERG is key to diagnosis as it shows the severe or complete loss of function of cone photoreceptors while rod photoreceptors are spared. Reported OCT phenotypes are extremely heterogenous, ranging from foveal hypoplasia to widespread ellipsoid zone loss and do not appear to show age-dependence nor structure–function correlation [26,27]. While it has been described that the disease can be characterized by OCT sequentially, the findings do not correlate with decline in function. In the affected NHPs, the fundi were noted to have a subtle maculopathy. On OCT imaging, foveal thinning was demonstrated with an intact ellipsoid zone. While color testing was difficult to conduct in monkeys, the ERG conducted on affected individuals showed normal rod responses but completely absent cone responses [19]. In this model, while the disease-causing mutation was identified, *PDE6C* mutations only account for 1% of clinical cases of achromatopsia in humans [28], with *CNGB3* and *CNGA3* accounting for the majority of cases. Hence, the applicability of this model in therapeutic development has to be further explored.

There may still be many naturally occurring IRDs in NHPs with shared pathogenic mechanisms with humans that remain undiscovered. However, the process of identifying these is challenging. For NHPs to manifest detectable behavioral changes due to visual impairment, most IRDs must have already undergone significant progression. In such circumstances, it may be difficult to characterize the IRD-specific phenotypic changes apart from widespread retinal degeneration. On the other hand, conducting an ophthalmic screen or genetic testing of all NHPs in a facility is resource-intensive and may not be cost-effective. Lastly, despite the genetic similarity, some naturally occurring mutations in the NHP genome may not be pathogenic in humans. In such circumstances, the clinical relevance of the disease model for studying IRDs in humans may be challenged.

## 5. IRD NHP Models Created Using Chemical and Laser Methods

To engineer viable NHP models for studying retinal degenerations, chemical and laser-based models that recapitulate retinal atrophy seen in end-stage IRDs have been created. Unfortunately, these models poorly capture pathogenesis of retinal degeneration in IRD. However, due to the relative ease for disease model generation, they are still used widely, particularly in the context of assessing cellular therapeutics.

The use of chemical methods for the creation of NHP retinal degeneration models was first reported by Shirai et al. in 2016 during a transplantation study of human embryonic stem-cell-derived retinal tissue. Cobalt chloride was administered at the subretinal level to induce full thickness outer nuclear layer (ONL) degeneration. Localized ONL degeneration was confirmed through retinal imaging and eventual histopathological confirmation, while loss of function was demonstrated via ERG. These changes were stable for up to 7 months [10]. This model has been used for the evaluation of other retinal cellular therapeutics, with variable outcomes [29]. Sodium iodate NHP models were also subsequently attempted [30]. While the mechanisms responsible for sodium-iodate-associated retinal cell death are not well-defined, ferroptosis has been suggested to be a possible mechanism [31]. The administration of intravenous sodium iodate was shown to be fatal for NHPs. However, intravitreal administration resulted in rapid and severe retinal degeneration [11]. Sodium nitroprusside, an anti-hypertensive drug, when administered subretinally, can also result in retinal degeneration via oxidative damage from nitric oxide release. Over a 28-day period, the subretinal SNP injections resulted in the disruption of retinal organisation coupled with involvement of the choroid. ERG also revealed functional deficits in areas of injury. The lesions were stable and persisted for up to 5 months [12].

Laser-induced models allow for the creation of focal retinal lesions through the adjustment of laser parameters, including the wavelength, power, spot size and exposure duration of the laser [10,13]. Initial studies suggested that longer wavelength lasers result in damage of the outer retina layers, while shorter wavelength lasers damage all retinal layers [8]. Other studies have combined femtosecond laser with adaptive optics to achieve focal photoreceptor ablation in NHPs [9]. The ability to selectively limit damage to the outer retina layer can be useful as RPE disruption compromises the blood–retina barrier and evokes a strong immune response, whereas damage to the inner retina impedes functional re-circuiting. The MicroPulse^®^ laser, which has also been utilized to create focal RPE lesions in porcine models [32] via the delivery of short repetitive pulses with intervals to allow the retina to cool down, is slowly being adapted for use in creating NHP models [33]. However, these laser models do not replicate the widespread damage to the outer retinal layer that is observed in humans and have limited utility in elucidating the pathogenic mechanisms underlying IRD.

## 6. Genetic Methods for IRD NHP Model Creation

Compared to chemical and laser-induced methods for IRD NHP model creation, genetic methods can better replicate the pathogenic process underlying IRD development. Manipulation of the NHP genome through knock-down, knock-in or knock-out of specific genes allows for the characterization of the phenotype–genotype relationship of IRD-associated genes. These can be achieved through gene editing or silencing using DNA or RNA-based tools. DNA-based tools, which usually target gene transcription, include the use of Zinc Finger Nucleases (ZFN), Transcription Activator-Like Effector Nucleases (TALEN) and Clustered Regularly Interspaced Short Palindromic Repeats (CRISPR) methods [34], while RNA-based tools, such as RNA interference and anti-sense oligonucleotides, and even more recent RNA editing methods [35], exist for targeting gene translation, although the effects are usually transient. Observing the natural history of IRDs and conducting long-term investigations of therapeutics in these models may be challenging. Hence, they have less utility in NHP IRD model creation and will not be discussed in this manuscript. In this segment, we will explain the mechanisms of DNA-based tools. Examples of NHP IRD models that have been created using these tools will also be discussed.

### 6.1. DNA-Based Methods

Gene editing is a form of genetic engineering where an organism’s DNA is edited through insertions or deletions. Gene editing technologies have been utilized to create NHP IRD models such as the CRISPR-Cas RP model and achromatopsia model. In the following segments, the mechanisms of gene editing technologies will be explained and the two NHP models will be discussed.

Non-homologous end joining (NHEJ) and homologous recombination (HR) are two major pathways for the repair of double-stranded breaks (DSB) in DNA that were discovered in the late 1900s. These processes are the basis of gene editing and can be exploited [36]. NHEJ modifies broken DNA ends and joins them up without factoring in any homology, resulting in unwanted sequence insertions or deletions. NHEJ is useful for gene knock-out. Meanwhile, HR relies on an undamaged DNA strand to guide the repair of the DSB, leading to the reformation of a sequence which closely resembles the original. By providing a synthetic DNA template strand, the HR mechanism can be exploited to correct any unwanted sequences, making it useful for editing specific DNA sequences.

While the initial generation of gene editing tools such as Zinc Finger Nucleases (ZFN) and Transcription Activator-Like Effector Nucleases (TALEN) were difficult to design, the emergence of simpler and more predictable CRISPR-based gene editing tools has revolutionized the field. Apart from therapeutic development, these tools have also accelerated the creation of animal models [37]. Table 2 provides a comparison of gene editing technologies. In the following segment, the mechanisms of these tools will be explained and the IRD NHP models that have been created using these tools will be discussed.

#### 6.1.1. Zinc Finger Nucleases (ZFN)/Transcription Activator-Like Effector Nucleases (TALEN)

ZFN and TALEN were conceptualized from studying the Fok1, Type II (S) restriction enzyme. As compared to other restriction enzymes, Fok1 is unique as it has separate DNA-cleavage and DNA-recognition domains. Most importantly, the DNA-cleavage domain has no specificity and can work independently as long as it is guided to a DNA strand [38].

In the 1990s, Chandrasegaran et al. showed that the DNA-cleavage could be redirected by substituting the natural DNA-recognition domain with zinc finger (ZF) domains, one of the most common DNA-binding domains in mammals [39,40]. By combining the Cys_2_His_2_ ZF, which can bind up to 30 amino acids [41], to Fok1, many different genetic sequences can be cleaved to allow recombination to occur [42]. In a ZFN, two ZFN proteins have to be created as Fok1 requires dimerization to function. Each ZF DNA binding domain can recognize 3 bps. By combining multiple ZFs together, a longer nucleotide sequence can be recognized. In general, ZFN are designed to demonstrate around 18–36 bp of specificity.

The next generation of gene editing tools to be developed were TALEN. In nature, TALE proteins activate plant genes to support the virulence of Xanthamonas, a plant pathogenic bacteria. The TALE protein is comprised of three domains: (1) amino-terminal with a transport signal, (2) DNA-binding domain made of 34 repeating sequences of amino acids arranged in tandem and (3) carboxyl-terminal with a nuclear localization signal and transcription activation domain [43,44]. The DNA-binding domain contains two hypervariable amino acids, which are known as the repeat variable diresidue. These determine the nucleotide-binding specificity of each repeat [45].

Similar to ZFNs, TALE proteins are fused to Fok1 restriction enzyme to form TALEN. Compared to ZFN, there are specific TALE proteins that recognize only 1 bp rather than 3 bp in the DNA-binding domain. The TALE proteins can be joined together to create a highly specific and modifiable tool to target almost any DNA sequence in the genome. With up to 30 to 40 BP of specificity, TALEN is, in theory, still the most specific gene-editing tool available [46].

Between the late 1990s and early 2002s, both ZFN and TALEN were rapidly adopted for targeted genetic engineering as they provided an effective method for gene knockout and gene editing. Apart from therapeutic applications, these technologies were also used to generate animal models of various diseases. In the IRD realm, ZFN and TALEN have been used to create small animal models such as mice [47] and zebrafish [48,49,50]. However, to date ZFN and TALEN have not been used to create IRD NHP models.

While TALEN provided far superior sequence specificity to ZFN, the use of both technologies still required the use of significant protein engineering methods, which was resource- and time-consuming. Furthermore, TALEN, while providing superior specificity, has a large protein size. Hence, choosing an appropriate vehicle for TALEN delivery has been challenging. These challenges may explain a lack of ZFN- and TALEN-based IRD NHP models.

#### 6.1.2. Clustered Regularly Interspaced Short Palindromic Repeats/CRISPR-Associated (CRISPR/Cas) Methods

CRISPR/Cas systems can be found in almost 90% of all bacteria, and archaea and provide significant immunity against viruses [51]. These systems are made of arrays of repeated sequences, interspersed by spacers, which are short, 20–50 bp long, non-repetitive DNA segments. The spacers are portions of the viral genome that are added to the CRISPR sequence during infection. These spacer arrays can be transcribed and eventually processed into small CRISPR RNA (crRNA) to recognize invading nucleic acids and mark them for eventual degradation [52,53]. In summary, CRISPR/Cas system works in three phases: (1) integration of spacer sequences; (2) processing of CRISPR locus transcript and maturation of crRNA; and (3) DNA or RNA interference [54].

In 2012, when Doudna and Charpentier et al. demonstrated the ability to program the Cas9 system from *Streptococcus pyogenes* to function as a RNA-guided DNA endonuclease [55], several groups, including Feng Zhang et al., continued modifying the system for application in mammalian cells [56,57]. Today’s CRISPR/Cas systems are extremely simple, with only a single guide RNA (sgRNA) and a Cas protein. The sgRNA is typically 98–100 bp long, with the 5′ end having a protospacer that recognizes the sequence of interest and the 3′ end having a transactivating cRNA (tracrRNA). With this structure, the sgRNA guides the Cas protein to the protospacer adjacent motif (PAM). PAM is a short DNA sequence (2–6 bp long) that follows the DNA region targeted for cleavage by the CRISPR/Cas system. It is generally 3–4 nucleotides downstream from the cut site. Once the Cas protein induces DSB, further gene repair mechanisms will then take place. As compared to ZFN and TALEN, CRISPR/Cas systems do not require complex protein engineering methods. Hence, the design of such systems is simpler and less resource-intensive, making it widely available to many life science laboratories (Table 1). Its simplicity has also revolutionized animal model development. To date, NHP models for achromatopsia and RP have been generated using CRISPR/Cas technology. These NHP models will be discussed in the subsequent paragraphs.

In a 2021 study by Li et al., the adeno-associated virus (AAV) serotype shH10 was used as a vector to deliver CRISPR/*Staphylococcus Aureus* Cas9 (SaCas9) to knock out the *RHO* gene in the rod photoreceptors of *Macaca mulatta* in vivo, with the aim of generating a macaque model of retinitis pigmentosa. sgRNAs targeting the first exon of the *RHO* gene were designed to achieve a high rate of complete gene knockout. SaCas9 was chosen over the conventional Streptococcus pyogenes Cas9 as the former is about 1 kb shorter and would therefore be able to fit within the packaging limitation of AAV (about 4.85 kb), allowing for the construction of both Cas9 and sgRNA into one AAV vector for high co-transduction rate. Each sgRNA under the control of the U6 promoter was individually cloned with SaCas9 under the control of the human synapsin I (hSyn) promoter to drive neuron-specific expression. In vitro, the cleavage efficiency of sgRNA was about 50%. In each test eye, three subretinal injections of AAV/ShH10-hSyn-SaCas9-U6-sgRNA1, 2, 3 plasmids were given, and about 10–20% of the retina was determined to be infected by AAV by immunohistochemistry. Significant indel-existing reads were found at the desired location, suggesting likely dysfunctional production of *RHO* proteins. Furthermore, no mutations were detected at potential predicted off-target loci. Morphological studies of the virus-infected Cas9-*RHO* retinae showed distinct photoreceptor degeneration, with reduced rhodopsin expression to ~47% that of control retinae, as well as reduced opsin (long-, mid- and short-wavelength) expression to ~27% that of control retinae, suggesting secondary loss of cone photoreceptors. There was complete loss of ONL in the macula after 8 months, indicating progressive photoreceptor degeneration. Furthermore, on FA, hyperfluorescent areas where virus was injected subretinally were seen, suggestive of leakage of retinal telangiectasia. On OCT, the ellipsoid zone was either disrupted or absent, while total retinal thickness and photoreceptor thickness of infected macula and periphery were observed to significantly decrease over time. On transmission electron microscopy, abnormal subcellular structures of infected photoreceptors were seen, with vacuolated mitochondria, shortened and disorganized rod discs, and strong cell apoptosis. In agreement with the morphological findings, ex vivo ERG testing showed significantly decreased photoresponse in infected areas compared to non-infected areas. Taken together, this study provided convincing evidence of the generation of an NHP RP model that closely mimicked class A RP disease in humans, with demonstrable loss of RHO protein, early rod photoreceptor degeneration, thinning retinae and reduced physiological functions [15].

In 2020, Lin et al. reported the use of AAV9 as a vector to deliver CRISPR/*Streptococcus pyogenes* Cas9 (SpCas9) to knock out the *CNGB3* gene in the cone photoreceptors of *Macaca fascicularis* in vivo, with the aim of generating a macaque model of achromatopsia. sgRNAs targeting exon 6 of the *CNGB3* gene were designed, and the sgRNA with the highest targeting efficiency was cloned into a AAV9 vector under the control of the U6 promoter. SpCas9 was chosen instead of the smaller SaCas9 as there were more SpCas9 protospacer adjacent motifs in exon 6, and it was driven by the small but less efficient elongation factor promoter in order to fit the AAV packaging limit. Unlike the *RHO* knockout study where a single Cas9-*RHO* shH10 vector was used, this study used two separate AAV9 vectors in view of the larger SpCas9 size—one for SpCas9 and the other for the sgRNA. A premix of both vectors was subsequently injected subretinally into three separate sites in each test eye, and about 12–14% of the retina was determined to be infected by both immunohistochemistry and single-cell sequencing of isolated cones. This partial knockout of the *CNGB3* gene was demonstrated to result in consistent reduction of mfERG response at D90 post-injection but not overall retinal function as measured by ffERG, suggesting cone dysfunction in central macula is consistent with an achromatopsia phenotype [14].

### 6.2. Challenges in Genetic Methods for Generation of IRD NHP Models

#### 6.2.1. Significant Resources Required for NHP Germline Editing

A key consideration prior to developing transgenic animal models is deciding on either a germline or somatic genome editing approach. Germline editing involves making genetic changes to reproductive cells such as sperm or eggs. The edited gene will be present in all cells of the eventual organism. Somatic editing involves editing the genome of targeted cells in the body. The edited gene will only be present in targeted cells. Traditional germline transgenic models were generated via the following methods: (1) DNA microinjection, (2) embryonic stem-cell mediated gene transfer and (3) viral-mediated gene transfer. However, these approaches have mainly been used for small animals such as mice instead of NHPs. An advantage that germline editing provides is that the genetic modifications can be passed down in subsequent generations, while somatic editing does not allow this. Although TALENs and CRISPR have been used for germline editing in NHPs, there are still no such IRD models available [58,59,60,61,62,63]. However, there are resource and ethical hurdles to NHP germline editing. Significant resources can be incurred due to the animal’s lengthy gestation, predominantly singleton births, large space requirements and required expertise involving handling NHP germline cells and breeding. Ethical concerns such as iatrogenic injury from procedures used to create the models and off-target effects leading to unintended phenotypic characteristics can have a negative impact on the welfare of these animals. Given the ease of utilizing gene editing tools, international discussion on relevant ethical policies are currently underway [64].

#### 6.2.2. Editing Efficiency

While many CRISPR/Cas systems have shown high genome-editing efficiency in-vitro, in-vivo results may differ drastically. In both the CRISPR/Cas IRD NHP models above, the systems produced less than a 20% gene editing rate in-vivo [14,15]. Multiple factors can influence the editing efficiency of gene editing systems. These include the DSB repair mechanisms, the design of DNA-recognition domains in ZFN, TALEN, sgRNA for CRISPR/Cas systems, and the method of delivery. Several methods of suppressing NHEJ and enhancing HDR have been published to increase gene-editing efficiencies [65,66]. Other strategies include increasing the injected dose or number of injection sites, or combining multiple sgRNAs in a single delivery vector [67].

#### 6.2.3. Delivery Methods

Delivery methods of gene editing mechanisms can be broadly classified into viral-mediated and non-viral mediated. The most well-known delivery method is the AAV vector. Other more common viral vectors include lentiviral, adenoviral and retroviral vectors. Two key considerations when choosing delivery methods are the transgene capacity and tissue tropism.

Tissue tropism eventually affects the transduction efficiency. Even within the serotypes of a particular viral vector, tissue tropism can vary. For instance, among AAV serotypes, AAV 2, 5, and 7–9 are capable of transducing photoreceptors, whereas almost every AAV serotype is capable of infecting the RPE, although several studies have offered contradictory results depending on the species tested, the route of vector delivery, and the health of target tissues [68,69,70]. AAV serotype 9 may be the most efficient vector to target both rod and cone photoreceptors and RPE cells via subretinal injection [71].

However, the restricted transgene capacity (4.5–5.0 kb) of AAV vectors remains a major limitation. To this end, lentiviral vectors, which have a cloning capacity of up to 10 kb, pose an attractive alternative as a single lentiviral vector can carry all the components of the CRISPR/SpCas9 system. Studies on lentiviral vector-mediated gene delivery to the eye have been well-reviewed [72,73,74], with many choosing to use the equine infectious anemia virus (EIAV) [75,76,77,78]. TALEN systems, depending on the design, may require vectors with even larger transgene capacities, while ZFN systems, with their small sizes, can be delivered quite easily.

#### 6.2.4. Off Target Effects

Off-target effects are non-specific, unintended genetic modifications to other areas of the genome that demonstrate similar but not identifical sequences. While in theory gene editing technologies are designed to be specific, in reality, off-target effects can occur in all 3 modalities [79,80]. Several factors affect the likelihood of off target effects, including the number of homolgous off-target sites [81] and degree of nuclease expression [82]. In particular, larger genomes such as the NHP genome are likely to have a greater number of homologous off-target sites, making this a significant consideration when designing CRISPR/Cas NHP models.

Several strategies have been undertaken to reduce off-target cleavage in CRISPR/Cas based systems. These include designing high-specificity mutant Cas9 nucleases [83,84,85] or mutant Cas9 ‘nickases’ that are only able to induce a single strand break [86], optimizing gRNAs by using truncated sgRNAs [87] and using various methods such as whole genome sequencing to detect off-target effects [88,89,90,91].

#### 6.2.5. Genetic Mosaicism

Genetic mosaicism is the presence of more than one genotype in an organism. It has been commonly reported in the generation of transgenic animals using CRISPR/Cas systems [92,93,94]. Mosaicism occurs when DNA replication happens before CRISPR-mediated genome editing. Mosaicism can significantly impact the process of creating a transgenic animal line as it complicates the intepretation of genotyped animals and reduces likelihood of a direct knockout generation. While the precise mechanisms underlying mosaicism have yet to be elucidated, prolonged expression of Cas9 mRNA is thought to increase mosaicism and off-targeting. Tu et al. showed that mosaic mutations in NHP embryos can be reduced by shortening the half-life of Cas9, which they achieved by tagging Cas9 with ubiquitin-proteasomal degradation signals [95]. Another group showed that complete target gene knockout was able to avoid a mosaic genotype [96]. By designing multiple adjacent sgRNAs spaced 10–200 bp apart to target a single key exon of each gene, and injecting the sgRNAs with Cas9 mRNA into monkey zygotes, they achieved 90–100% efficiency of gene knockout and did not detect any off-target mutations on whole-genome sequencing of the positive samples.

## 7. Conclusions

Given the significant shared similarities between NHPs and humans, NHP IRD models have the potential to provide insight into the pathogenic processes involved. In turn, this can significantly de-risk and accelerate the therapeutic development of gene therapy. Although models created using genetic methods can better recapitulate the pathogenesis of IRDs compared to laser and chemical methods, there are concerns of cost and efficacy. As improvements are made to current genetic editing technology, especially in terms of editing efficiency and reducing off-target editing, it is likely that more NHP IRD models will emerge over the subsequent years.

## Figures and Tables

**Table 1 genes-13-00344-t001:** Summary of non-human primate models of inherited retinal diseases.

Reference	Species	Inherited Retinal Disease	Genotype	Mechanism of Model	Comments
**Naturally Occurring Non-Human Primate Models**					
Discovery of a Cynomolgus Monkey Family with Retinitis Pigmentosa (Ikeda et al., 2018) [6]	*Macaca fascicularis*	Retinitis pigmentosa	**Not identified**	Naturally occurring	One identified primate had severe parafoveal degeneration (loss of outer retinal layers) and complete loss of ERG responses; Related primate had retinal degeneration limited to the peripheries with almost unrecordable dark-adapted ERG and extremely reduced light-adapted ERG; No conclusive mutation identified from genotyping limits the application of this NHP model.
Bardet–Biedl Syndrome in rhesus macaques: A nonhuman primate model of retinitis pigmentosa (Peterson et al., 2019) [7]	*Macaca mulatta*	Bardet–Biedl Syndrome (BBS) Retinitis Pigmentosa	**BBS7** (c.160delG, p.Ala54GlnfsTer18)	Naturally occurring	Identified primates had severe macular degeneration noted structurally on AF, OCT and histology (loss of both retina and RPE) and functionally on ERG; Primates also manifested other aspects of BBS including renal impairment and hypogonadism; Identified mutation in a gene associated with BBS in humans; Animals were identified only after significant progression of disease. Further studies on progression of disease are required.
A non-human primate model of inherited retinal disease (Moshiri et al., 2019) [5]	*Macaca mulatta*	Achromatopsia	**PDE6C** Homozygous R565Q missense mutation (c.1694G>A, p.Arg565Gln)	Naturally occurring	Relatively normal looking retina but hyper autofluorescence at the fovea on AF imaging, similar to humans, was observed; Unrecordable cone responses on ERG, a key diagnostic feature of human achromatopsia; Causative gene identified is only responsible for 1% of all achromatopsia cases in humans.
**Iatrogenic Non-Human Primate Disease Models (Chemical and Laser-Induced Methods)**					
Focal damage to macaque photoreceptors produces persistent visual loss (Strazzeri et al., 2014) [8]	Not stated	Retinal Degeneration	Gene agnostic	Laser-induced	Laser used: Coherent Novus Omni 647 nm laser, single shot mode, continuous wave; Settings: power (100–260 mW), spot size (200–250 µm) and pulse duration (10–200 ms); Focal loss of outer retinal layers with relative sparing of inner retinal layers; Lesions recovered over 2 months; Undesirable effects: RPE damage in addition to photoreceptors, strong immune response, and adhesion of retina to RPE (impeding infiltration of transplanted donor cells).
Localized Photoreceptor Ablation Using Femtosecond Pulses Focused With Adaptive Optics (Dhakal et al., 2020) [9]	*Macaca fascicularis* and *Macaca mulatta*	Retinal Degeneration	Gene agnostic	Laser-induced	Laser used: Ti:Sapphire femtosecond laser; Settings: power (50–210 mW) and pulse duration (106–335 ms); Two-photon adaptive optics scanning light ophthalmoscope used to deliver ultrafast laser exposures; Selective photoreceptor ablation without disruption of RPE or inner retina.
Transplantation of human embryonic stem cell-derived retinal tissue in two primate models of retinal degeneration (Shirai et al., 2016) [10]	*Macaca fascicularis* and *Macaca mulatta*	Retinal Degeneration	Gene agnostic	Drug-induced Laser-induced	**Drug-Induced** Subretinal injection of cobalt chloride (0.3 mg/mL); Focal loss of outer nuclear layer followed by inner retinal layers demonstrated over period of 7 months on OCT, fluorescein angiography and histology; Corresponding negative focal ERG at each injury site; Narrow effective dose window; Variable efficacy in inducing retinal degeneration. **Laser-Induced** Laser used: PASCAL 577 nm laser, 5 × 5 mode; Settings: power (110–175 mW), spot size (100 µm) anduration (15 ms); Focal lesions appeared 4 days after intervention but became less evident after 2 months; No choroidal neovascularisation for up to 2 months after injury; Uneven damage to photoreceptor and RPE regions—occasional insufficient photoreceptor damage.
Establishment of Retinal Degeneration Model in Rat and Monkey by Intravitreal Injection of Sodium Iodate (Ou et al., 2018) [11]	*Macaca fascicularis*	Retinal Degeneration	Gene agnostic	Drug-induced	Intravitreal injection of sodium iodate; Narrow effective dose window; Rapid and widespread lesion development.
Establishment of a Rapid Lesion-Controllable Retinal Degeneration Monkey Model for Preclinical Stem Cell Therapy (Gao et al., 2020) [12]	*Macaca fascicularis*	Retinal Degeneration	Gene agnostic	Drug-induced	Subretinal injection of sodium nitroprusside (0.1 mM); Focal loss of outer retinal layers (outer plexiform, outer nuclear layers and RPE) noted on histology and OCT; Reduced amplitude on mfERG over damaged region; Persistent damage lasting for at least 7 months; Narrow effective dose window.
Localized Structural and Functional Deficits in a Nonhuman Primate Model of Outer Retinal Atrophy (Liu et al., 2021) [13]	*Macaca fascicularis*	Retinal Degeneration	Gene agnostic	Laser-induced	Laser used: PurePoint 532 nm laser, single shot mode; High power settings: power (250 mW), spot size (50 µm) and duration (200 ms); Lower power settings: power (150mW), spot size (50 µm) and duration (200 ms); Focal loss of outer retinal, RPE and choriocapillaris noted on OCT, OCTA, fluorescein angiography and histology. Localised mfERG dysfunction noted in first month after lesion induction; Loss of choriocapillaris may limit use of model in transplantation studies due to reduced graft viability; Does not mimic pathogenic mechanisms.
**Non-Human Primate Models Created Using Genetic Methods**					
Generation of nonhuman primate model of cone dysfunction through in situ AAV-mediated *CNGB3* ablation (Lin et al., 2020) [14]	*Macaca fascicularis*	Achromatopsia	*CNGB3*	**Gene knockout** CRISPR-Cas9 knockout Dual AAV9 vector Somatic Subretinal injection	Used Streptococcus pyogenes (spCas9); Delivered via dual AAV9 vector; 12–14% targeting efficiency demonstrated on immunohistochemistry and single-cell transcriptomic analysis; Model demonstrated reduction of mfERG response at D90 post-injection but no overall reduction of ffERG response, consistent with cone dysfunction in the central macula.
Generation of non-human primate retinitis pigmentosa model by in situ knockout of *RHO* in rhesus macaque retina (Li et al., 2021) [15]	*Macaca mulatta*	Retinitis pigmentosa	*RHO*	**Gene knockout** CRISPR-Cas9 knockout Somatic Single AAV9 vector	Used Staphylococcus aureus Cas9 (saCas9) instead due to smaller size for packaging (~1 kb shorter); 10–20% targeting efficiency demonstrated on immunohistochemistry and single-cell transcriptomic analysis.

**Table 2 genes-13-00344-t002:** Comparison of gene-editing tools (ZFN, TALEN and CRISPR/Cas).

Characteristic	ZFN	TALEN	CRISPR/Cas
**Design and Construction**	Difficult, requires protein engineering	Difficult, require protein engineering	Simple
**Endonuclease**	Fok1	Fok1	Cas9
**DNA Specificity**	18–36 bp	30–40 bp	22 bp
**Delivery**	Easy to deliver with viral vectors due to small size	Challenging to deliver with AAV vectors due to large size	Challenging to deliver with AAV vectors due to large size
**Multiplexing**	Difficult	Difficult	Very feasible
**Off target editing**	High	Low	Moderate

## Data Availability

No data was reported in this study.

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
