# Peer review of "Developing Non-Human Primate Models of Inherited Retinal Diseases"

_genes, 2022, doi:10.3390/genes13020344_

Round 1
Reviewer 1 Report
Thanks for your working hard for collecting nearly all the previous model of inherited retinal disease.
Therefore, we could easily realize the development of the study field.
Thanks.
Reviewer 2 Report
The review by Seah and co-workers presents the state of the art on NHP models for IRD. The review is well written and collects information of different naturally occurring or experimentally induced models in NHP. They also discuss pros and cons of new methodologies based on genome editing.
I have few suggestions to improve the review:
- Sentence at line 123: while the mouse lacks a EYS homolog, I am not sure that this is the case for ABCA4 and RDH5. In fact, knock-out models are available for these two genes but, maybe, the models do not perfectly represent the disease. I suggest revision of this sentence
- Line 129: “non human-IRD causative gene…” did they mean “no human-IRD causative gene”?
- Line 263: the statement “reformation of the original” is not always correct. In fact, this method is commonly used to generate Knock-in with changes of 1 or more bases. Please, revise this sentence.
- Paragraph lines 283-287 should be revised because how it is now written can be interpreted as that one ZNF can bind 18-36bp. This is not correct, because the recognition to the specific sequence derives from an array of ZNFs and each ZNF recognizes only 3 bases. Please revise.
- In the presentation of the CRISPR/Cas9 system a mention to the fact that the DNA cleavage happens at sequences called PAM, specific for each Cas, should be explained. Here it appears that there are no rules for the design of a gRNA.
Minor comments:
- Line 101: “diseases” could be “disease”
- Line 105: I suggest to delete “similar”
- Line 257: “(HR) were 2” should be “(HR) are 2”
- Line 300: “BP” should be “bp”
